# Adherence to Prescribed Acamprosate in Alcohol Dependence and 1-Year Morbidities and Mortality: Utilizing a Data Linkage Methodology

**DOI:** 10.3390/jcm10102102

**Published:** 2021-05-13

**Authors:** Serenella Tolomeo, Alex Baldacchino

**Affiliations:** 1Department of Psychology, National University of Singapore, Singapore 117572, Singapore; 2Division of Population and Behavioural Sciences, St Andrews University Medical School, St Andrews KY16 9TF, UK

**Keywords:** adherence, alcohol, acamprosate, mortality, morbidities, data linkage

## Abstract

Objectives: We tested the hypothesis that poor adherence is associated with a greater risk of alcohol-caused mortality and morbidities within the first year of discontinuing this medication. Materials and Methods: A retrospective cohort study of 3319 individuals who received acamprosate in the East of Scotland in a 10-year period was conducted using a health informatics approach with record linkage of dispensing data, hospital utilization (SMR) and General Register Office of Scotland (GROS) data. The primary outcome was adherence between one to six months after initiating acamprosate medication. The secondary outcome was all-cause morbidities and mortality. Results: Of the total 3319 individuals identified, a good adherence index of >80% was found in 59% of those prescribed acamprosate after three months and 6% after six months. There were significant linear trends of poorer adherence with increased risk of alcohol-caused mortality (Hazard Ratio, HR 1.2), medical morbidities especially neoplasm (HR 4.1) and poisoning (HR 1.4), and psychiatric morbidities especially stress (HR 35.1), psychotic (HR 5.6) and neurotic disorders, and directly alcohol induced conditions (7.4 HR) after adjustment for other factors within a one-year period of initiation of acamprosate treatment. Discussion and Conclusions: Further exploratory studies using this digitalized approach should be encouraged in order to capture role of compliance to acamprosate and other types of medication that are known to reduce relapse into alcohol dependence and its direct relationship to mortality and morbidities in this population.

## 1. Background and Significance

Alcohol dependence (AD) is a chronic relapsing disorder, and the 3.3 million deaths every year directly attributed to harmful use of alcohol make it a major challenge to public health. It contributes substantially to the global burden of disease with 5.3% of all deaths and 5% of disability-adjusted life years [1]. The health impact is most severe amongst young adults, where alcohol attributes to approximately 25% of deaths in the age group 20–39 years [2].

Although many benefit from treatment, a low occurrence of treatment-seeking is a common denominator for the majority of people suffering from alcohol use disorder (AUD), a relapsing disorder characterized by compulsive alcohol use, loss of control over alcohol intake, and a negative emotional state when not using [3]. Findings from the National Epidemiologic Survey on Alcohol and Related Conditions (NESARC) show that in the USA only 14.6% of individuals who met lifetime criteria for an AUD reported having received alcohol treatment [4,5]. This denotes that a large group of individuals fail to seek help, are not offered help, or meet other limitations in accessing treatment to recover from their AUD [4,5]. In a systematic review by May et al. (2019), three barriers proved to be prominent: shame and stigma, lack of perception of treatment need, and the paradox of both need for and fear of giving up drinking [6].

Acamprosate is an effective and well-tolerated medication in conjunction with psychosocial and behavioral treatment programs [7,8] for individuals with severe forms of alcohol use disorder (alcohol dependence). According to the United Kingdom’s National Institute of Clinical Excellence (NICE) guidelines [9] the use of acamprosate is recommended as a first-line treatment for at least six months after successful detoxification from alcohol provided that the individual is tested for normal liver function prior to initiation of this medication. Thompson and colleagues conducted an open cohort data linkage study design of 1257 alcohol-dependent individuals on a prescription of acamprosate as identified from the UK Clinical Practice Research Database (CPRD) between 1990 and 2013 [10]. Results from this study indicate that ”prescribing persistence” (a record of repeat prescription within 90 days of the expected end date of their last prescription) for those receiving acamprosate was 27.7% at 6 months, 13.7% at 12 months and 7.5% at 18 months after initiating acamprosate [10]. Many of the individuals in the Thompson et al. study never received a repeat prescription, with a median duration of therapy of 2.10 months (95% CI (Confidence Interval) 1.87 to 2.53).

## 2. Objectives

Here, we report for the first time the relationship between adherence to acamprosate and the risk of all-cause morbidities and mortality after 1 year using a health informatics approach with record linkage of routine healthcare data.

## 3. Materials and Methods

### 3.1. Setting

We identified people resident in the eastern part of Scotland who were registered with a general practitioner and were prescribed and dispensed acamprosate in a 10-year period. Every individual registered with a general practitioner in Scotland is assigned a 10 digit unique patient identifier, the Community Health Index (CHI) number used in all correspondence with the NHS (National Health Service) in Scotland [11]. This number, which includes the person’s date of birth, allows linkage of health-related datasets, providing a unique resource combining information on dispensed prescriptions with detailed clinical data at the level of the individual patient. This study population represents all individuals attending the publicly run UK National Health Service in the Tayside and Fife regions of Scotland (population approximately 800,000), which delivers any medical and psychosocial interventions for free to all age group experiencing any communicable or non-communicable pathology that needs treated and/or managed by primary and/or secondary care systems [12].

### 3.2. Individuals on Acamprosate

We collected data on age, sex, and postcode for each individual who was dispensed an acamprosate prescription from the Community Health Index number register, which covers the entire population registered with a general practitioner. We used the number to link healthcare records to all community dispensed prescription and to standard morbidity records (SMRs) [13] records for admission to medical and psychiatric inpatient units. In addition, we established linkages to the General Register Office of Scotland (GROS) [14] mortality data, and to laboratory datasets relating to urine testing for opioids and other illicit drugs.

We used census data to calculate Scottish Index of Multiple Deprivation (SIMD) scores and subsequent categories for social deprivation for each individual based on their home postcode [15]. Each record of a prescription for acamprosate contains details on the individual’s CHI number, and all redeemed prescriptions are recorded for reimbursement purposes. We did not include individuals who were prescribed more than one course of acamprosate treatment and/or were exhibiting polydrug dependence.

We used liver function tests (alanine aminotransferase (ALT), aspartate aminotransferase (AST) and gamma-glutamyl transferase (γGT)) to determine whether the individuals who died were biologically homogeneous at start of acamprosate treatment to the individuals who were still alive. This was deemed as an adequate objective proxy confirmation of biological homogeneity.

### 3.3. Ethical, Information, and Research Governance

We followed standard operating procedures at the Health Informatics Centre (HIC) Services, University of Dundee, to ensure anonymity of the dataset (CA/FB/HIC SOP). The Tayside Committee on Medical Research Ethics (GN13AL058) and NHS Fife and Tayside Caldicott Guardians approved the study. The University of Dundee was the sponsor for this study (30 December 2012).

### 3.4. Calculation of Adherence

Adherence indices were derived for acamprosate from the dispensing database. The information on dosage, frequency, length of prescription, and number of tablets dispensed indicated the maximum possible length of drug coverage for each prescription. We calculated the proportion of the total number of days of drug coverage (PDC) and divided by the total number of days of follow-up in the study and expressed it as a percentage or ”adherence index” [16,17,18].

We used this measure with the aim to examine each day in the specified period to determine if the patient had acamprosate coverage. To provide a direct comparison with previous studies, we calculated the adherence index from the first prescription to six months of treatment. Based on the existing literature [19,20], we used these operational definitions:We categorized individuals as having good adherence to acamprosate if the adherence index was ≥80%;We categorized individuals as having low adherence to acamprosate if the adherence index was <80%.

### 3.5. Datasets and Data Linkage

Primary Care Pharmacy dispensing data [21], GROS data [14], and medical admissions (SMR01), psychiatric admissions (SMR04), and Accident and Emergency (A&E) attendance data [13] were obtained for each individual of the study population from the corresponding datasets. Each dataset contained the CHI number, which facilitated record linkage, along with one primary and five other possible diagnosis codes (from the International Classification of Diseases Tenth Revision—ICD-10). For this study, we linked dispensing data with (1) GROS deaths and (2) morbidities. We also used the number of different types of prescriptions as a proxy measure of comorbidity.

For the mortality and morbidity modelling we categorized the individual primary ICD10 codes for complication of interest as:Cardiovascular: e.g., chronic ischemic heart disease, cardiomyopathy, heart failure;Cerebrovascular: e.g., intracerebral hemorrhage, cerebral infarction, cerebral aneurysm;Neoplasm: e.g., neoplasm of the pancreas, neoplasm of the liver;Poisoning: e.g., poisoning by narcotics, poisoning by topical agents, poisoning by nonopioid analgesics;Respiratory: e.g., pulmonary disease, pneumonia, bronchitis, respiratory failure;Mental and behavioral disorder: e.g., anxiety, obsessive compulsive disorder, stress, mood disorder, schizophrenia, withdrawal state with delirium;Hepatic: e.g., cirrhosis, liver disease, chronic hepatitis;Gastrointestinal: e.g., gastro-esophageal reflux disease, gastrointestinal hemorrhage, gastroenteritis, and colitis.

We considered the following arbitrary categories to further explore the possible relationship of alcohol in the identified morbidities and mortality within 1 year of prescribing acamprosate.

For medical admissions:(1)Not direct: cardiovascular, cerebrovascular, neoplasm, poisoning and respiratory;(2)Direct: alcohol-related and unspecified alcohol issues, hepatic, gastrointestinal and alcohol-related causes on the certificate of death.

For psychiatric admissions:(1)Not direct: acute stress, mood disorder, anxiety and panic attack;(2)Direct: mental and behavioral disorders due to use of alcohol and alcohol induced conditions (e.g., withdrawal, harmful use, dependence and hallucinations).

### 3.6. Statistical Methods

Data were described as number (percentage) of individuals for categorical variables and mean (SD) for continuous variables. Continuous variables that did not follow a normal distribution were tested with the Shapiro-Wilks test for skewness and reported as median and interquartile range. We reported χ2 tests for the distribution of the population.

We used survival regression models to predict mortality outcomes within five years since the initiation of acamprosate. The first prescription for acamprosate was designated the index date for follow-up. We used Cox proportional hazards model to help express the relationship between mortality and age, gender, social deprivation, co-morbidity, and polydrug pharmacy as covariates. We included these covariates in the multivariate model used if we deemed them to be of clinical significance and/or if they had a univariable *p*-value below 0.05.

We assessed the proportional hazards assumption by using trend tests of the Schoenfeld residuals. Those that failed the assumption or were deemed to be time-dependent were entered as continuous time-dependent covariates. Outcomes were mortality and morbidities within a 1-year period.

In a community-based, non-randomized study, many known and unknown factors may determine who receives acamprosate, which can potentially bias results. A propensity score estimating the probability of an individual receiving acamprosate was then calculated using a logistic regression model of receipt of acamprosate (yes/no) for age, gender, social deprivation, co-morbidity and polydrug pharmacy. This was added to the survival models to adjust for propensity to receive acamprosate and therefore reduce bias within the survival analytic model. We used SPSS version 21 (IBM, Armonk, NY, USA) and SAS version 9.2 (SAS Institute, Cary, NC, USA) for all statistical analyses.

## 4. Results

### 4.1. Descriptive Statistics

Data were obtained on 3319 individuals who were prescribed acamprosate. Table 1 shows the descriptive statistics for the study population. The mean age was 48.41 years old.

The number of individuals who had redeemed their acamprosate prescription in the six months since initiation of treatment was 71% after the first month, 67% after the second month, and then less than 50% after the third month. At the end of the 6 months, only 6% of the original population redeemed their acamprosate prescription.

The biochemistry characteristics of the individuals who died within one year are shown in Appendix A showing biological homogeneity between high- and low-adherence individuals before acamprosate was prescribed. Additionally, the prescribing practice of this cohort of non- adherent population had not changed from that of the adherent populations as all NHS patients are all managed under local, regional, and national treatment standards and guidelines for alcohol dependence.

### 4.2. Morbidities

#### 4.2.1. Medical Admissions (SMR01)

A multivariate model was created to investigate the risks of hospital admissions within the first year of starting acamprosate. We associated this with type of morbidity as direct (31%) or not alcohol-related (69%) as described in the methods section. The results showed that low-adherence individuals were at significantly risk of attending hospital (*p* = 0.01), and especially people with neoplasm and poisoning were at significantly risk of attending hospitals (*p* < 0.001 and *p* < 0.01). The results of the model are shown in Table 2.

#### 4.2.2. Psychiatric Admissions (SMR04)

A multivariate model was created to investigate the risks of attending psychiatric hospitals within the first year of starting acamprosate. The results showed that low-adherence individuals were at significantly risk of admissions to psychiatric hospitals (*p* = 0.02). People with more hazards were individuals who had acute stress (*p* < 0.001), alcohol induced conditions (*p* < 0.01), schizophrenia (*p* < 0.001), mental and behavioral disorder secondary to the use of alcohol (*p* < 0.01), or anxiety (*p* < 0.001). The results of the model are shown in Table 3.

#### 4.2.3. Accident and Emergency (A&E) Admissions

A multivariate model showed that low-adherence individuals were at significantly risk of attending A&E within the first year of starting acamprosate (Hazard Ratio, HR: 1.0; *p* < 0.05).

#### 4.2.4. Other Medication as Proxy Measure of Morbidities

Results of multivariate modelling show that prescription of other medications (e.g., antidepressants), used as an additional proxy measure of comorbidities, did not show any significant association (*p* < 0.089) with low adherence.

### 4.3. Cause of Death

Of the 3319 individuals identified, 252 (7.6%) died within one year during the study. The characteristics of the cohort are shown in Appendix A.

The principal cause of death was: “mental and behavioral disorder due to use of alcohol” followed by “alcohol liver disease”. Table 4 summarizes the other principal causes of death.

### 4.4. Multivariable Associations between Covariates and Causes of Mortality

A multivariate model showed that low-adherence individuals were at significant risk of death compared to individuals who were high-adherence (*p* < 0.001). Older individuals were also at a significantly increased risk of death compared to younger individuals (*p* < 0.001) (Appendix A).

## 5. Discussion

Randomized controlled trials and systematic reviews have shown that acamprosate has minimal effect compared to placebo in reducing lapse/relapse of alcohol use and subsequent increase in the percentage of alcohol-free days [22,23,24]. However, the findings of our study show that increased adherence to acamprosate is associated with reduced risk of death and morbidities related to alcohol-related causes. To our knowledge, this is the largest study of adherence to acamprosate in the general population utilizing a health informatics cross linkage digitalized approach. For example, a systematic review and meta-analysis showed that only eight studies have investigated mortality in relation to acamprosate. The total number of patients studied was 2677 participants, which included people with comorbid depression. It was concluded that the strength of evidence of acamprosate in relation with mortality was poor due to the small sample size [25].

We report three main findings. First, at six months after starting acamprosate, only 6% of individuals had followed the recommended minimal 6-month treatment. Second, individuals who were categorized as having low adherence were at a significantly higher risk of death and morbidities within 12 months of starting acamprosate. Third, causes of these deaths and morbidities were highly associated with alcohol-related disorders. This is consistent with previous studies with naltrexone treatment, which is also prescribed for alcohol dependence, reporting that there was a significant relationship between good adherence to medication and successful treatment outcomes [22]. Additionally, consistent with our findings, another study predicted that if acamprosate treatment coverage is increased to 40%, this would result in a significant decrease of alcohol-related mortality and morbidities in one year [26,27].

However, these results highlight important clinical implications. Many of the research models for interventions to improve medication adherence are both time- and staff-intensive and thus beyond the capacity of an individual clinician to implement [28]. This study significantly adds to the evidence base for the effective implementation of the prescription of acamprosate for people with alcohol dependence who wish to remain abstinent. Currently, relapse to drinking alcohol is common with only a limited range of medications available as an adjunct to psychosocial programs. Whilst many individuals benefit from them, not everyone does so; therefore, understanding and maximizing the delivery of these effective treatments is important. Devoting time to addressing adherence during the course of treatment is crucial to minimize added risk of mortality as a result of non-adherence.

There are several limitations to this study. We restricted our cohort to a specific timeline to avoid the possible influence that changing treatment philosophies and procedures might have in determining pharmacological provision and subsequent treatment outcomes. However, as stated earlier, the treatment guidelines and subsequent clinical practice has not changed throughout the study period with NHS Tayside and Fife Alcohol Services utilizing the same approved standard operating procedures and protocols for the treatment of alcohol dependence with acamprosate. A second limitation is that we identified people registered with acamprosate coverage prescriptions because it was important to use routinely collected data to provide high precision for the related outcome. This may represent a weakness of representativeness, although it covered both urban and rural populations. However, we could have missed a proportion of people who were not registered with GPs or were changing residential status to outside Scotland. However, it is expected that such cases are minimal.

Thirdly, this study did not have a control group (a) with another AD population prescribed another medication focused on relapse prevention such as disulfiram (Antabuse^®^) or naltrexone; (b) with an AD population prescribed nalmefene focused on a reduction in binge alcohol consumption; or (c) with an AD population not on any medication, actively heavily drinking and attending for example accident and emergency services. This would have helped to better contextualize the results [29]. Additionally, one important point to highlight is that we did not have any proxy measures to indicate any health-related issues prior to collection of data that might have influenced susceptibility to increased mortality and morbidities during the 1-year follow up period. However, we used biological proxy measures to come close to understanding this potential confounder by determining liver function tests before initiation of acamprosate prescription. We could have also collected dosage of benzodiazepine used during the alcohol detoxification schedule prior to acamprosate prescription, as this is a proxy measure of severity of alcohol withdrawal and severity of alcohol dependence. Additionally, we explored secondary outcomes for non-alcohol-related medical and psychiatric comorbidities, hospital admissions, and co-prescription during the first year after initiating acamprosate. This would improve face validity of the results achieved utilizing disparate but closely linked datasets.

## 6. Conclusions

Further exploratory studies using this digitalized approach should be encouraged in order to capture role of compliance to acamprosate and other types of medication that are known to reduce relapse into alcohol dependence and its direct relationship to mortality and morbidities in this population.

## Figures and Tables

**Table 1 jcm-10-02102-t001:** Characteristics of individuals medicated with acamprosate.

	Total
Number	3319
Ethnicity (white)	3319 (100%)
Mean age (SD) in years	48.41 (11.88)
Males	2047 (61%)
Females	1272 (39%)
SIMD	
1 most deprived	529 (16.0%)
2	471 (14.0%)
3	430 (12.6%)
4	374 (10.6%)
5	237 (8.0%)
6+ least deprived	1176 (35.5%)
Total number of deaths within the 10-year study period	617
Total number of Individuals who died within 1 year of starting acamprosate	252

SIMD = Scottish Index of Multiple Deprivation; SD = standard deviation.

**Table 2 jcm-10-02102-t002:** Multivariate association between covariates and all causes registered in SMR01 (medical admissions) within the first year of starting acamprosate.

Predictor	HR	95% CI	*p*-Value
Overall attendance at hospitals			0.001
Low adherence	1.001	1.0–1.002	0.01
**Reasons**			
Cardiovascular	1.049	0.765–1.44	0.802
Cerebrovascular	2.27	0.705–7.311	0.249
Neoplasm	4.104	2.277–7.075	<0.001
Poisoning	1.406	1.175–1.684	<0.01
Respiratory	1.428	0.99–2.062	0.11
Mental and behavior disorder	0.919	0.642–1.317	0.7
Hepatic	1.053	0.802–1.384	0.754
Gastrointestinal	1.375	1.0	1.892

HR = hazard ratio; CI = confidence interval, *p* = significance value of <0.05.

**Table 3 jcm-10-02102-t003:** Multivariate association between covariates and all causes of SMR04 (psychiatric admissions) within the first year of starting acamprosate.

Predictor	HR	95% CI	*p*-Value
Overall attendance at psychiatric hospitals			<0.001
Low adherence	3.00	3.0–3.002	0.02
**Reasons**			
Acute stress	35.11	14.53–86.47	<0.001
Mood disorder	2.463	1.11–5.65	0.063
Alcohol induced conditions: withdrawal, harmful use, dependence, hallucinations	3.734	1.79–7.8	<0.01
Schizophrenia	5.580	3.1–10.1	<0.001
Mental and behavioral disorder due to use of alcohol	7.412	2.43–22.67	<0.01
Anxiety disorder, OCD, panic attack	647.3	181.93–2303.19	<0.001

OCD = Obsessive Compulsive Disorder.

**Table 4 jcm-10-02102-t004:** Causes of death from General Registry Office of Scotland (GROS) death certificates (*n* = 252) within a 1-year period.

Causes	ICD-10 Code	Examples	Number of Deaths within a 1-Year Period (%)
Alcohol liver disease	K70–K77	cirrhosis liver disease chronic hepatitis	30 (33)
Mental and behavioral disorder due to use of alcohol	F10	alcohol withdrawal state with delirium mental and behavioral disorder due to use of alcohol and epilepsy	53 (57)
Gastrointestinal	K40–K63	gastro-esophageal reflux gastrointestinal hemorrhage gastroenteritis colitis	1 (2)
Cardiovascular	I00–I69	chronic ischemic heart cardiomyopathy heart failure	12 (5)
Neoplasm	C00–C72	neoplasm of the pancreas neoplasm of the liver	7 (3)

ICD-10 = International Statistical Classification of Disease and Related Health Problems Version 10; K70–K77 = diseases of the liver; F10 = alcohol-related disorders; K40–K63 = diseases of the digestive system; IOO–I69 = diseases of the circulatory system; C00–C72 = neoplasm.

## Data Availability

Data is not available as analysis was conducted through a specified time limited period utilizing analytic facilities within the safe haven within HIC as per standard operating procedure.

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
