# Peer review of "Adherence to Prescribed Acamprosate in Alcohol Dependence and 1-Year Morbidities and Mortality: Utilizing a Data Linkage Methodology"

_jcm, 2021, doi:10.3390/jcm10102102_

Round 1

Reviewer 1 Report

This paper presents the test of the hypothesis that poor adherence to the treatment with Acamprosate is associated with a greater risk of alcohol-caused mortality and morbidities within the first year of discontinuing this medication. The paper reads well and the methods is well described. The recommendation is to accept the paper after taking into account the next specific comment.

I suggest the authors to put the clinical implication before the limitations and to end the article reporting the conclusions referred in the abstract

Reviewer 2 Report

4.2. Morbidities 4.2.1. Medical Admissions (SMR01) A multivariate model was created to investigate the risks of hospital admissions within the first year of starting acamprosate. We associated this with type of morbidity as direct 31%) or not alcohol-related (69%) as described in the method section. The results showed that low adherent individuals were at significantrisk of attending hospital (p=0.01),and especially people with neoplasm and poisoning were at significantrisk of attending hospitals (p<0.001 and p<0.01).What results for high adherent subjects? May you insert this information?4.2.2. Psychiatric Admissions (SMR04)What results for high adherent subjects? May you insert this information?Have you matched AUD subjects with and without psychiatric comorbidity?Page 3, first paragraph “We used liver function tests ...”: The kidneys are primarily responsible for theeliminationofacamprosate! Acamprosate is contraindicated in patients with severe renal impairment and gastrointestinal adverse events may prevent use or limit dose maximization.DiscussionsRaw 1-2: Not all people agree with this affirmation: for example, “ Palpacuer, C., Duprez, R., Huneau, A., Locher, C., Boussageon, R., Laviolle, B., Naudet, F., 2018. Pharmacologically controlled drinking in the treatment of alcohol dependence or alcohol use disorders: a systematic review with direct and network meta-analyses on nalmefene, naltrexone, acamprosate, baclofen and topiramate. Addiction 113, 220237”.

Reviewer 3 Report

Dear authors

this paper is well written with a very good and new  methodology and should be published. I am working a lot on the long term course of alcoholdependence and on medications for relapse prevention and we published in 2021 abook with an overview on this topic:

Otto-Michael Lesch, Henriette Walter, Christian Wetschka, Michie N. Hesselbrock, Victor Hesselbrock and Samuel Pombo(second Edition): Alcohol and Tobacco Medical and Sociological Aspects of Use, Abuse and Addiction, Springer Verlag

In this book there are nearly 1000 publications to this topic and if you like I can send it by mail, Following these results I would propose some small improvements for the paper:

1)Introduction: ,,, is a progressive and chronic....... please only 24 % are progressive 33% are episodic and , please cut the tterm progressiv in this paper

2) you assessed also withdrawal, if you assessed also the severity, it would be interesting to know if accamprosate was givweb ib these patients who developed severe withdrawal, because following the basic and clinical literature to accamprosate in animal studies it was proved only in withdrawal models and our clinical data showa that is is working esspecially in alcohol depence with severe withdrawal or withdrawal seizures, if you don`t have this data it should be mentioned in the discussion

3) Please in the discussion it have to be mentioned the we treat with different medications different mechanism : Disulfiram, nalmefene before drinking, accamprosate to stay sober ,naltrexone to reduce the amount of drinking and sodium oxy  bate for a substitution therapy (e.g.:

Author Response

Please see that attachment
